# Childhood Rotavirus Infection Associated with Temperature and Particulate Matter 2.5 µm: A Retrospective Cohort Study

**DOI:** 10.3390/ijerph182312570

**Published:** 2021-11-29

**Authors:** Hui-Chen Tseng, Fung-Chang Sung, Chih-Hsin Mou, Chao W. Chen, Shan P. Tsai, Dennis P. H. Hsieh, Chung-Yen Lu, Pei-Chun Chen, Ya-Ling Tzeng

**Affiliations:** 1School of Nursing and Graduate Institute of Nursing, China Medical University, Taichung 406, Taiwan; htseng@mail.cmu.edu.tw; 2Department of Nursing, China Medical University Hospital, Taichung 404, Taiwan; 3Management Office for Health Data, China Medical University Hospital, Taichung 404, Taiwan; fcsung1008@yahoo.com (F.-C.S.); b8507006@gmail.com (C.-H.M.); 4Department of Health Services Administration, China Medical University College of Public Health, Taichung 404, Taiwan; 5Department of Food Nutrition and Health Biotechnology, Asia University, Taichung 413, Taiwan; 6University of Maryland Global Campus, Adelphi, MD 20783, USA; chaochen25@comcast.net; 7School of Public Health, Texas A&M University, College Station, TX 77843, USA; shantsai99@gmail.com; 8Department of Environmental Toxicology, University of California, Davis, CA 95616, USA; dphsiehcmu@gmail.com; 9Department of Sport and Health Management, Da-Yeh University, Changhua 515, Taiwan; u100030082@cmu.edu.tw; 10Department of Public Health, China Medical University College of Public Health, Taichung 406, Taiwan; eliz0118@gmail.com

**Keywords:** ambient temperature, fine particulate matter, interaction, rotavirus infection, seasonality

## Abstract

No study has ever investigated how ambient temperature and PM_2.5_ mediate rotavirus infection (RvI) in children. We used insurance claims data from Taiwan in 2006–2012 to evaluate the RvI characteristics in children aged ≤ 9. The RvI incidence rates were higher in colder months, reaching the highest in March (117.0/100 days), and then declining to the lowest in July (29.2/100 days). The age–sex-specific average incident cases were all higher in boys than in girls. Stratified analysis by temperature (<20, 20–24, and ≥25 °C) and PM_2.5_ (<17.5, 17.5–31.4, 31.5–41.9, and ≥42.0 μg/m^3^) showed that the highest incidence was 16.4/100 days at average temperatures of <20 °C and PM_2.5_ of 31.5–41.9 μg/m^3^, with Poisson regression analysis estimating an adjusted relative risk (aRR) of 1.26 (95% confidence interval (CI) = 1.11–1.43), compared to the incidence at the reference condition (<20 °C and PM_2.5_ < 17.5 μg/m^3^). As the temperature increased, the incident RvI cases reduced to 4.84 cases/100 days (aRR = 0.40, 95% CI = 0.35–0.45) when it was >25 °C with PM_2.5_ < 17.5 μg/m^3^, or to 9.84/100 days (aRR = 0.81, 95% CI = 0.77–0.93) when it was >25 °C with PM_2.5_ > 42 μg/m^3^. The seasonal RvI is associated with frequent indoor personal contact among children in the cold months. The association with PM_2.5_ could be an alternative assessment due to temperature inversion.

## 1. Introduction

Childhood rotavirus infection (RvI) is the leading cause of diarrhea in children worldwide and is most common in infants and children younger than 5 years old. The virus caused diarrhea with near 454,000–705,000 deaths in children in 2000–2004 worldwide [1,2]. The disease has been greatly reduced by the introduction of vaccines. Prior to the vaccination, in 2006, rotavirus infection was also the most prevalent cause of severe diarrhea for U.S. children, with 55,000–70,000 hospitalizations but only 20–60 deaths annually [3]. Rotavirus positivity reduced from 25.6% in the pre-vaccine period of 2000–2006 to 6.1% in the post-vaccine period of 2007–2018, with a reduction of 76.2 % [4]. Despite vaccination, the virus was still responsible for 125,800 deaths from diarrhea in young children in 2016 worldwide [5,6]. Reinfections with rotavirus are also common throughout life [7]. The mortality risk has remained much greater in developing countries than in developed countries. Over 80% of deaths occurred in sub-Saharan Africa. The disease is also a major cause of hospitalizations for gastroenteritis in South African children [8]. In Southeast Asia, the disease accounted for 40.78% of all diarrheal diseases in children in the last decade [9].

Rotaviruses are transmitted mainly through person-to-person contact, primarily by the fecal–oral route [10]. Lower socioeconomic status is associated with poor sanitation conditions and hygiene practices, and healthcare facilities are a major factor leading to higher episodes of and mortality from the infection [5,9].

However, the transmission of the virus has been associated with ambient environmental conditions with distinct seasonality [9,11,12,13,14,15,16,17]. A meta-analysis of 26 studies showed that RvI in the tropics is likely higher in cool and dry months [14]. Studies in South Asia and Southeastern Asia found that the infection was more common in colder months [9,13]. Studies in Hangzhou, China, and Hong Kong also showed higher infection risks in winter than in summer [15,16]. A recent study analyzed 15,991 diarrhea specimens at a medical center in Taiwan from 2013 to 2018 and found near 70% of rotavirus-positive samples were detected in January–March [17]. However, an earlier study in Bangladesh and a recent study in Peru showed an opposite seasonal phenomenon: RvI was associated with higher temperatures [18,19].

The evidence shows that infectious rotavirus particles could be identified from a variety of surfaces and objects, as well as the hands of caregivers, in daycares [10,12]. Some studies also suspected that rotaviruses may be airborne and spread on fomites and environmental surfaces [12,16,20,21,22]. A study in Beijing identified abundant inhalable soil-associated microorganisms nonpathogenic to humans in particulate matter samples collected in cold months, with bacterial species accounting for over 80% and with 0.1% viral reads, but no rotavirus was reported [20]. A hospital-based study in Hangzhou reported that children with rotavirus infection were positively associated with haze, including particular matter, CO, SO_2_, and NO_2_ [16]. However, the study failed to elaborate on how the positive relationship between the infection and haze is inversely associated with the climate conditions.

No population-based study has evaluated how the risk of RvI associated with climate condition interacts with air quality, or clarified a causal relationship. To characterize relationships, we used data from Taiwan’s insurance claims to determine how RvI’s seasonality was also related to air quality. We focused our analyses on the risk of infection based on the average monthly temperature and the level of PM_2.5_ in each temperature stratum in children aged 9 and under.

## 2. Materials and Methods

### 2.1. Data Source and Study Population 

This study used the National Health Insurance Research Database (NHIRD) to research children aged < 10, obtained from the Ministry of Health and Welfare of Taiwan, and data of climate conditions and air quality, obtained from the Taiwan Environmental Protection Administration (EPA). The insurance database provides claims data of outpatient and inpatient care for half of all children in 1996–2012, consisting of information on treatments, medication, and costs. All personal identifications in the claims data were recoded for linkage and protection of the privacy of insured residents. The diseases were coded using the International Statistical Classification of Diseases 9th Revision (ICD-9).

The Environmental Protection Administration has established 76 air quality monitoring stations since 1993 on the island, with a total area of 36,197 km^2^, to monitor the hourly meteorological conditions and ambient air pollutants. For this study, hourly measured values of temperature and the PM_2.5_ level were used. Because PM_2.5_ monitoring was not conducted before 2006, insurance claims and environmental data from 2006 to 2012 were used in this study.

### 2.2. Ethics Statement

Personal identifications in the data files were recoded with surrogate numbers to ensure personal privacy. The research ethics committee requirement for patient consent was waived. China Medical University and Hospital approved the study (CRREC-103-018). This study was carried out in accordance with the Code of Ethics of the Declaration of Helsinki.

### 2.3. Statistical Analysis

The data analysis first calculated and plotted the monthly average incidence of RvI rate (per 100 days) by the monthly average temperature from January to December for the period of 2006–2012. We further calculated and plotted age-specific (1–5 and 6–9 years) average incident cases per 100 days by average temperature level (<15, 15–19, 20–24, 25–29, and ≥30 °C) for boys and girls. The overall average incident cases per 100 days were calculated by stratified levels of temperature (<15, 15–19, 20–24, 25–29, and ≥30 °C), PM_2.5_ (<17.5, 17.5–31.4, 31.5–41.9, and ≥42.0 μg/m^3^), and parental income for insurance premiums (NTD <250,000, 250,000–299,999, and >300,000). The Poisson regression analysis estimated the relative risk (RR) and 95% confidence interval (CI) of RvI associated with temperature, PM_2.5_, and parental income. We further evaluated how temperature and PM_2.5_ mediated the RvI risks by estimating the incidence and RRs of RvI by PM_2.5_ stratum (<17.5, 17.5–31.4, 31.5–41.9, and ≥42.0 μg/m^3^) in each temperature stratum (<20, 20–24, and ≥25 °C), using the incidence at <20 °C and PM_2.5_ < 17.5 μg/m^3^ as a reference. Multivariable analysis was used to estimate the adjusted relative risk (aRR) of infection after controlling for sex, age, and parental income. 

## 3. Results

### 3.1. RvI Incidence Inversely Related to Average Temperature

The monthly average temperature ranged from 16.4 °C in January to 29.2 °C in July in Taiwan during the period of 2006–2012 (Figure 1). The monthly average incidence rate of childhood RvI was inversely associated with the monthly mean temperatures, with a peak incidence of 117.0/100 days in March, which declined to 29.2/100 days in July. 

The incident cases per 100 days were over three times higher in children aged ≤ 5 than those aged 6–9 and were higher in boys than in girls, with similar trends by the mean temperature (Figure 2). 

### 3.2. Incidence and Relative Risk of RvI by Temperature, PM_2.5_, and Parental Income

The RvI incidence by temperature was the highest (71.6 cases/100 days) at the mean temperature of 15–19 °C, with an aRR of 2.25 (95% CI = 2.11–2.41) compared with the incidence of 29.2/100 days at ≥30 °C (Table 1). The incidence rate increased with the PM_2.5_ level, from 30.8/100 days at <17.5 μg/m^3^ to 63.0/100 days at ≥42.0 μg/m^3^. Children from high-income families were four times more likely to be diagnosed with RvI than children from lower-income families (108.8 versus 27.1 per 100 days). 

Table 2 shows the trends of RvI incidence rates associated with temperature, PM_2.5_, and parental income were similar for children aged ≤ 5 years and 6–9 years. Children who were 6–9 years old from higher-income families were at higher risk of being detected with RvI than children aged ≤ 5 years old. 

### 3.3. RvI Associated with Interaction between Temperature and PM_2.5_

Table 3 shows that the infection rate increased with the PM_2.5_ concentration in each temperature stratum. In the temperature stratum of <20 °C, the aRR of RvI was 1.39 (95% CI = 1.24–1.57) at the PM_2.5_ level of ≥42.0 μg/m^3^ compared to that of <17.5 μg/m^3^. In the temperature stratum of 20–24 °C, the aRRs of RvI were less than 1.0, associating with PM_2.5_ levels of <17.5, 17.5–31.4, and 31.5–41.9 μg/m^3^. The RvI incidence reduced to less than 10 per 100 days, with aRRs ranging from 0.40 to 0.81, being significant at the 5% level when the temperature was ≥25 °C.

## 4. Discussion

Previous studies on the seasonality of RvI in children have focused mainly on climate conditions which often follow a distinctive cycle, with mainly a winter peak in temperate regions [15,16,17,22]. Limited studies have also suspected the relationship of airborne transmission [12,16,20,21,22,23,24]. Our study used stratification analysis to evaluate how childhood RvI was associated with temperature and PM_2.5_ as indicators. We first demonstrated a strong inverse relationship between the monthly infection rates and temperatures, or a positive relationship between the monthly infection rates and PM_2.5_. We were able to use the large amount of data to evaluate the infection by stratified levels of temperature and PM_2.5_, showing an inverse relation between these two factors in the infection risk. 

Our seasonal relationship is consistent with patterns elsewhere with a temperate climate in particular, such as in Hangzhou [16], Hong Kong [15], and the Mediterranean island Mallorca of Spain [22]. However, meteorological conditions vary among areas, and temperature is a particularly distinctive factor associated with infection in the cold months. The incidence peaks in March when it ranges from 15 to 19 °C in Taiwan; in January when it ranges from 15 to 20 °C in Hong Kong; and in January–February when it is 9 °C in Mallorca; it is also high in January–March and November–December in Hangzhou when it is below 10 °C. The inverse relationship between childhood RvI and temperature could also be observed in a high mountain city, Kathmandu, Nepal [25].

Regardless of suspected airborne transmission [12,16,20,21,22,23,24], no clear relationship with air pollution was previously addressed. No study has ever detected the virus in aerosol, except experimental studies [26]. The eight-site MAL-ED cohort study reported that rotavirus infection in infants is associated with all nine hydrometeorological variables in low-income and middle-income countries [21]. The transmission is mediated by waterborne and airborne dispersal, viral survival, and host factors, but temperature remains the major mediator engaging in the transmission mechanisms of all factors. No air pollutant data were presented. The Hangzhou study reported that RvI in children increased with mean concentrations of PM_2.5_ and PM_10_. [16] However, the inverse relationship between RvI and temperature was much stronger than the positive relationship between RvI and air pollutants. Approximately 80% of RvI cases in Hangzhou were children younger than 18 months. These young children were not likely exposed to the ambient air pollutants, but no indoor pollution was reported. In Hangzhou, concentrations of air pollutants appear to be inversely related to the ambient temperature. This phenomenon is closely aligned with that of Taiwan; the PM_2.5_ level is higher on cold days. Therefore, RvI associated with air pollution is likely a surrogate relationship because of temperature inversion [27,28,29]. 

Our study presents a data analysis with the advantage of not only stratifying both temperature and PM_2.5_ but also providing a valuable means to further evaluate how RvI is associated with these two factors. Data analysis, by stratifying these two factors, showed that the RvI risk decreased with temperature and increased with PM_2.5_ (Table 2). The further estimated incident cases of RvI per 100 days by PM_2.5_ stratum in each temperature stratum showed that the infection risk was mainly temperature related, particularly when it was <20 °C (Table 3). The higher temperature exhibited a protective relationship, especially when it was at 25 °C and higher, even at the highest PM_2.5_ level of ≥42 μg/m^3^.

However, neither temperature nor pollutants are the causal factor of rotavirus transmission. These are factors that influence human behavior and activity. The transmission is indirectly associated with both of these factors. Rotavirus is a well-known, highly contagious virus transmitted mainly by the fecal–oral route, either person to person or through fomites in the environment [6,10,12]. In the present study, temperature was likely a stronger factor than PM_2.5_ in mediating the transmission. In our study, the lower monthly average temperature ranged from 16.9 °C in January to 19.5 °C in March, with the RvI rate peaking in March. In Taiwan, the winter break for school children usually falls in January and February for one month. All daycare facilities are also closed for the Chinese lunar new year, which also falls in February in most years. Rotavirus spread can easily increase among family members, especially among infants and young children. The spring semester generally starts in the final weeks of February, and the temperature remains lower with more rainy days in March. These weather conditions lead young children to have more indoor gatherings in places with increased person-to-person contact, such as daycares and schools. The infection declined with the increasing monthly temperature to the lowest rate during the summer break from mid-June to mid-September when there were fewer crowd gatherings, with minimum person-to-person contact among children.

It is likely that a similar transmission mechanism can be shared by people in other temperate areas, where the infection is higher in colder months. The hospitalization patterns for rotavirus gastroenteritis in Hong Kong [15] and the island of Mallorca [22], and for outpatient rotavirus detection in Hangzhou [16], are similar to the infection pattern in Taiwan, with a stronger relation to temperature than to other factors. Children in these regions may spend more time indoors and stay together when it is cold, increasing contact with infected individuals or contaminated materials. With monthly mean temperatures ranging from 7.90 to 26.1 °C in Nepal, infection risk is the highest when it is less than 10 °C [25]. Poorer hygienic practices in cold and overcrowded living conditions, such as not washing hands often, facilitate elevated rotavirus transmission.

It is also interesting to note that the risk of RvI was much higher in children from higher-income families in this study. Parents with higher incomes were more likely to seek healthcare for their children, which increased the detection of RvI. In Taiwan, high-income parents were more likely to send younger children to daycares and 6–9-year-old children to after school programs. Both can lead to more indoor gatherings. RvI might be thus increased.

This study has the advantage and strength of using large population-based data to evaluate how RvI is associated with ambient conditions. The monthly analysis and stratification analyses by both temperature and PM_2.5_ enabled us to delineate that the ambient conditions associated with RvI could result from human activities associated with the ambient conditions. However, there are some limitations to this study. First, at the average daily ambient temperature of 25 °C and higher, the aRR of RvI increased with the PM_2.5_ level from 0.40 to 0.81 but remained at a significant protective status compared to the condition with an average temperature of <20 °C and average PM_2.5_ of <17.5 μg/m^3^ (Table 3). In Taiwan, the EPA issues air pollution warnings and advises residents to stay indoors due to the pollution severity. We suspect that this increasing trend of RvI risk is associated with these warnings. Children at daycares and schools were more likely to be advised to stay indoors in response to the pollution warning, which likely increased the spread of rotavirus. However, information on pollution warnings was not available in the data. Second, we also suspected that temperature inversion is more likely to occur when it is cold, leading to the accumulation of PM_2.5_. Information on the inversion phenomenon was also unavailable to prove this hypothesis. Third, our data analysis did not evaluate RvI risks associated with rainfall, humidity, and geographical health disparity, which might lead to some residual confounding. Children are more likely to stay indoors on rainy days and are therefore at a higher RvI risk. Fourth, to date, the insurance program in Taiwan does not cover the cost of rotavirus vaccination for children. Information on vaccination was not available in the claims data. It is likely that our study might mainly represent the time before the rotavirus vaccination. 

## 5. Conclusions

This study demonstrated not only the well-known seasonality of RvI with a higher risk during the cold months but also a positive relationship with PM_2.5_. However, we consider these to be indirect relationships. The increased infection is likely associated with frequent indoor gatherings among children on colder days. The risk declines to the lowest in the hot months as children are on their summer break, meaning the person-to-person spread of the virus is likely reduced. PM_2.5_ levels are inversely associated with temperature inversion in Taiwan. The role of PM_2.5_ might not seem to be a priority, but whether there is an actual effect relationship requires further investigation.

## Figures and Tables

**Figure 1 ijerph-18-12570-f001:**
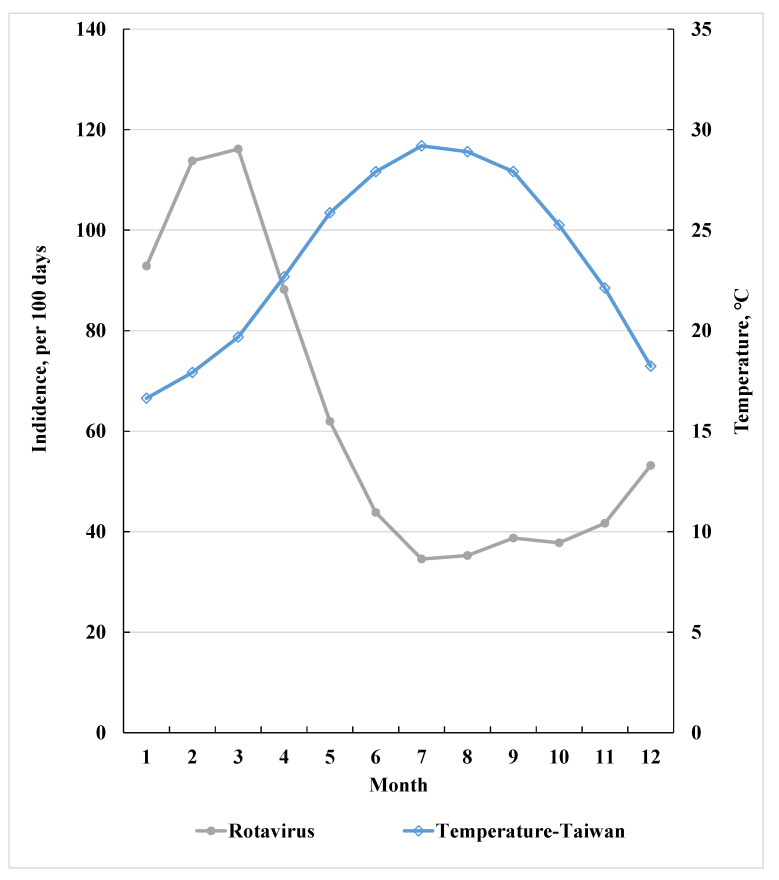
Relationship between monthly mean childhood rotavirus infection and average temperature in 2006–2012 in Taiwan.

**Figure 2 ijerph-18-12570-f002:**
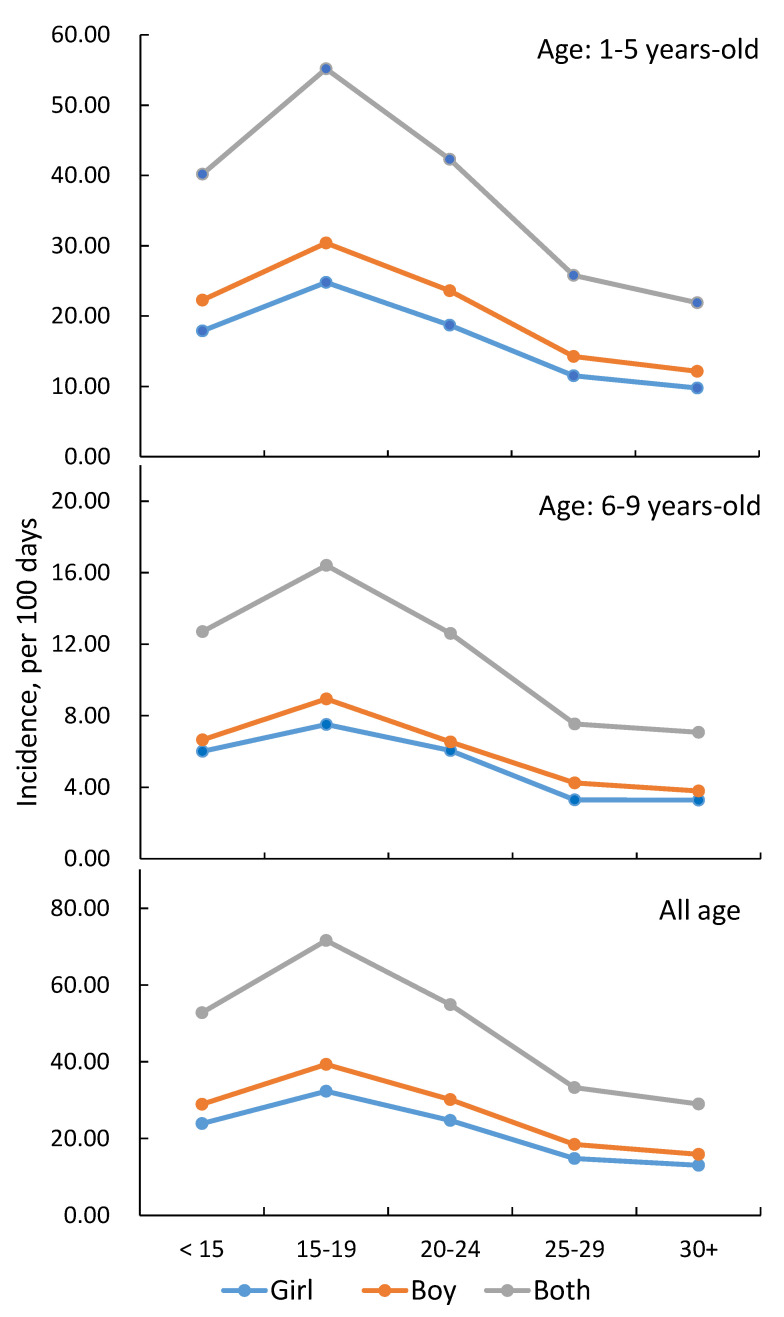
Childhood rotavirus infection cases per 100 days by age and average temperature (°C) for boys and girls in 2006–2012 in Taiwan.

**Table 1 ijerph-18-12570-t001:** Multivariable Poisson regression analysis estimated adjusted relative risk of rotavirus infection by temperature, PM_2.5_, and parental annual income.

Variable	Infection, *n*/100 Days	Relative Risk ^1^ (95% Confidence Interval)
Average daily temperature, °C		
<15	52.8	1.74 (1.61–1.88)
15–19	71.6	2.25 (2.11–2.41)
20–24	54.9	1.72 (1.61–1.84)
25–29	33.3	1.20 (1.12–1.28)
≥30	29.2	Ref.
Average daily PM_2.5_, µg/m^3^		
<17.5	30.8	Ref.
17.5–31.4	45.8	1.34 (1.29–1.38)
31.5–41.9	52.3	1.52 (1.46–1.58)
≥42.0	63.0	1.85 (1.78–1.92)
Parental income, NTD		
<250,000	27.1	Ref.
250,000–299,999	63.0	2.40 (2.33–2.46)
≥300,000	108.8	4.07 (3.95–4.20)

PM_2.5_, particular matter 2.5 μm^3^. ^1^ Multivariable analysis including sex, age, temperature, PM_2.5_, and parental income.

**Table 2 ijerph-18-12570-t002:** Multivariable Poisson regression analysis estimated relative risk of rotavirus infection by temperature, PM_2.5_, parental annual income, and age.

Variable	Adjusted Relative Risk (95% Confidence Interval) ^1^	
	≤5 Years	6–9 Years
Average temperature, °C		
<15	1.74 (1.59–1.89)	1.75 (1.50–2.03)
15–19	2.29 (2.12–2.47)	2.15 (1.88–2.46)
20–24	1.74 (1.62–1.88)	1.63 (0.42–1.87)
25–29	1.22 (1.13–1.32)	1.13 (0.99–1.30)
≥30	Ref.	Ref.
Average PM_2.5_, µg/m^3^		
<17.5	Ref.	Ref.
17.5–31.4	1.35 (1.30–1.41)	1.29 (1.20–1.38)
31.5–41.9	1.52 (1.45–1.59)	1.51 (1.39–1.63)
≥42.0	1.85 (1.78–1.93)	1.85 (1.72–1.99)
Parental income, NTD		
<250,000	Ref.	Ref.
250,000–299,999	2.21 (2.14–2.27)	3.21 (3.03–3.40)
≥300,000	3.75 (3.62–3.88)	5.50 (5.15–5.88)

^1^ Multivariable analysis including sex, age, temperature, PM_2.5_, and parental income.

**Table 3 ijerph-18-12570-t003:** Childhood rotavirus infection rate per 100 days by average daily PM_2.5_ level within each average temperature level and Poisson regression analysis estimated relative risk of the infection.

Temperature °C	PM_2.5_, μg/m^3^	Rate	Relative Risk Crude	(95% CI) Adjusted ^1^
<20	<17.5	14.8	Ref.	Ref.
	17.5–31.4	14.8	1.00 (0.88–1.12)	1.03 (0.92–1.17)
	31.5–41.9	16.4	1.11 (0.97–1.26)	1.26 (1.11–1.43)
	≥42.0	15.9	1.07 (0.96–1.21)	1.39 (1.24–1.57)
20–24	<17.5	7.81	0.53 (0.45–0.62)	0.59 (0.50–0.69)
	17.5–31.4	10.7	0.72 (0.64–0.83)	0.77 (0.68–0.88)
	31.5–41.9	12.0	0.81 (0.70–0.93)	0.91 (0.79–1.05)
	≥42.0	16.2	1.09 (0.98–1.23)	1.32 (1.18–1.48)
≥25	<17.5	4.84	0.33 (0.29–0.37)	0.40 (0.35–0.45)
	17.5–31.4	8.32	0.56 (0.50–0.63)	0.61 (0.54–0.68)
	31.5–41.9	8.95	0.60 (0.53–0.69)	0.70 (0.61–0.80)
	≥42.0	9.84	0.66 (0.58–0.76)	0.81 (0.77–0.93)

^1^ Adjusted for sex, age, and parental income.

## Data Availability

The raw insurance claims data in this study can be purchased from the Health and Welfare Data Science Center (HWDSC) after receiving IRB approval. Duplication of the insurance claims data is forbidden, while the ambient air pollution monitoring data are accessible from the EPA website.

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
