# Peer review of "Childhood Rotavirus Infection Associated with Temperature and Particulate Matter 2.5 µm: A Retrospective Cohort Study"

_ijerph, 2021, doi:10.3390/ijerph182312570_

Round 1

Reviewer 1 Report

This is a retrospective cohort study, based on health insurance statistics for rotavirus disease, on the correlation between rotavirus infection and ambient temperature and air pollution measured by presence of particulate mutter smaller than 2.5 µg in the air.

The study found, like many previous studies, an inverse correlation between the incidence of rotavirus and the temperature; i.e. more rotavirus in older months. While this finding is not new it is nicely documented graphically in Fig. 1. The novelty may be the climate zone of Taiwan. More drastic changes in RV incidence are seen in countries with colder climate.

The correlation between RV and small particular matter in the air as a new observation. Not surprisingly there was more RV when the quality of air was poorer. The authors consider this to be an indirect effect, and not a causal one, which may well be correct.

A surprising observation is that there was in Taiwan a considerable amount of RV in older children, i.e. 6-9 year-olds. This is probably because of the nature of the material. The authors should expand on this finding and explain how such cases end up in the Health Insurance data base. Can it be dissected to hospital treated and other cases?

The findings should also be put in perspective by explaining that the study represents time before RV vaccination. Universal RV vaccination will not only lower the total number of cases but also shift the peak incidence to spring, which means in warmer period.

Finally, the authors should find another word to replace “Mediated” in the title. Mediated does not fit well in direct effect.

Author Response

Response to Reviewer 1 Comments

Re: ijerph-1446247

New title: Childhood Rotavirus Infection Associated with Temperature and Particulate Matter 2.5 µm: A Retrospective Cohort Study

Open Review

Point 1: This is a retrospective cohort study, based on health insurance statistics for rotavirus disease, on the correlation between rotavirus infection and ambient temperature and air pollution measured by presence of particulate mutter smaller than 2.5 µg in the air.

Response 1: Thank you for the comment. Yes, we evaluated the RV associated with both ambient temperature and PM2.5, attempting to explain the association is associated with human behaviors and activities.

Children are more likely to stay indoors when it is cold. Children spending time together indoors increase the RV infection among them because it is transmitted mainly through person-to-person contacts primarily by the fecal-oral route [10]. The ambient PM2.5 level is high when it is cold. But, children are indoors and less likely expose to ambient PMs.

Point 2: The study found, like many previous studies, an inverse correlation between the incidence of rotavirus and the temperature; i.e. more rotavirus in colder months. While this finding is not new it is nicely documented graphically in Fig. 1. The novelty may be the climate zone of Taiwan. More drastic changes in RV incidence are seen in countries with colder climate.

Response 2: Yes, thank you for the comment.

Point 3: The correlation between RV and small particular matter in the air as a new observation. Not surprisingly there was more RV when the quality of air was poorer. The authors consider this to be an indirect effect, and not a causal one, which may well be correct.

Response 3: Yes, thank you for the comment.

Point 4: A surprising observation is that there was in Taiwan a considerable amount of RV in older children, i.e. 6-9 year-olds. This is probably because of the nature of the material. The authors should expand on this finding and explain how such cases end up in the Health Insurance data base. Can it be dissected to hospital treated and other cases?

Response 4: Thanks for the inspirational comment. Yes, RV infection is mainly prevalent among children 5 years old and younger, but not limited. Older children can be infected also with a lower infection rate. Adults can be infected as well, but rare (Anderson EJ, Weber SG. Rotavirus infection in adults. Lancet Infect Dis. 2004;4(2):91-99.)                          Our data in Figure 2 and Table 2 show that the infection patterns by temperature are similar between children of 1-5 years old and 6-9 years old. The incidence was approximately 3 folds higher in the 1-5 years old than in the 6-9 years old children. All cases were identified from claims data most from outpatient records.

Point 5: The findings should also be put in perspective by explaining that the study represents time before RV vaccination. Universal RV vaccination will not only lower the total number of cases but also shift the peak incidence to spring, which means in warmer period.

Response 5: Thanks for the inspirational comment. Yes, our study might mainly represent time before RV vaccination.

Both pentavalent (Rotateq, RV5) and monovalent (Rotarix, RV1) are available in Taiwan. Unfortunately, the insurance has not yet covered the cost. Therefore, information on the rotavirus vaccination information was not available in the data we use. This is another study limitation we added in our revision:

“Fourth, the insurance program in Taiwan has not yet covered the cost of rotavirus vaccination for children. The information on the vaccination was not available in the claims data. It is likely that our study might mainly represent time before rotavirus vaccination.” (Please see lines 271-274 in the revision)  

Point 6: Finally, the authors should find another word to replace “Mediated” in the title. Mediated does not fit well in direct effect.

Response 6: Yes, thank you for the inspirational suggestion. We have a brain storm.

We would like to change the old title “Childhood Rotavirus Infection is Mediated by Temperature and Particulate Matter 2.5 µm: A Retrospective Cohort Study” into:

“Childhood Rotavirus Infection Associated with Temperature and Particulate Matter 2.5 µm: A Retrospective Cohort Study”

Reviewer 2 Report

This is an interesting and well presented study.

Author Response

Response to Reviewer 2 Comments

Re: ijerph-1446247

New title: Childhood Rotavirus Infection Associated with Temperature and Particulate Matter 2.5 µm: A Retrospective Cohort Study

Open Review

Authors’ response

Points 1: This is an interesting and well presented study.

Response: Thank you very much for the comment.

Reviewer 3 Report

This manuscript used clinical data of rotavirus infection in children from 2006-2012 in Taiwan to evaluate the risk of RvI associated with ambient temperature and PM2.5. The results showed a strong inverse relationship between the monthly infection rates and temperatures, or a positive relationship between the monthly infection rates and PM2.5.

This is an interesting clinical study. It is well known that rotavirus infection occurred more frequently during cold season. The results of this study were consistent with previous publications. But the relation of RvI and PM2.5 is kind of new information and these data should be useful for RvI epidemiology study.

I have some minor concerns:

  1. Most time, the low temperature and high PM2.5 come together, it is hard to tell higher RvI is associated with cold weather or high PM2.5.
  2. The data also showed the relation between RvI and parental income. Would you please explain it in the discussion? Like why higher income families have more chance of RvI.
  3. Table 2. Please check if the first section needs to move the data down by one line.

Author Response

Response to Reviewer 3 Comments

Re: ijerph-1446247

New title: Childhood Rotavirus Infection Associated with Temperature and Particulate Matter 2.5 µm: A Retrospective Cohort Study

Open Review

Comment and authors’ response

Point 1: This manuscript used clinical data of rotavirus infection in children from 2006-2012 in Taiwan to evaluate the risk of RvI associated with ambient temperature and PM2.5. The results showed a strong inverse relationship between the monthly infection rates and temperatures, or a positive relationship between the monthly infection rates and PM2.5.

Response 1: Yes, thank you for the comment.

Point 2: This is an interesting clinical study. It is well known that rotavirus infection occurred more frequently during cold season. The results of this study were consistent with previous publications. But the relation of RvI and PM2.5 is kind of new information and these data should be useful for RvI epidemiology study.

Response 2: Thank you very much for the comment. Yes, the relationship with PM is interesting, requiring more attention and study.

Point 3: I have some minor concerns:

  1. Most time, the low temperature and high PM2.5 come together, it is hard to tell higher RvI is associated with cold weather or high PM2.5.

Response 3.1: It is more likely associated with temperature. When it is cold, children are more likely stay indoors. RvI is increased as children spend time together indoors because it is transmitted mainly through person-to-person contacts primarily by the fecal-oral route [10]. The ambient PM2.5 level is high when it is cold. But, children are in indoors and less likely expose to ambient PMs.

  1. The data also showed the relation between RvI and parental income. Would you please explain it in the discussion? Like why higher income families have more chance of RvI.

Response 3.2: Thank you for the inspirational comment. We have discussed this issue in the revision: It is also interesting to note in this study that the risk of RvI was much higher in children from higher income families. Parents of higher incomes were more likely to seek health care for children, which increased the detection of RvI. In Taiwan, high-income parents were more likely sending younger children to day cares and 6-9 years old children to attend after school programs. Both leaded to more indoor gatherings. The RvI might be thus increased.”            

  1. Table 2. Please check if the first section needs to move the data down by one line.

Response 3.3: Thank you for the reminder. We have moved the data down by one line.